# Ultrasonic Deep Brain Neuromodulation in Acute Disorders of Consciousness: A Proof-of-Concept

**DOI:** 10.3390/brainsci12040428

**Published:** 2022-03-23

**Authors:** Josh A. Cain, Norman M. Spivak, John P. Coetzee, Julia S. Crone, Micah A. Johnson, Evan S. Lutkenhoff, Courtney Real, Manuel Buitrago-Blanco, Paul M. Vespa, Caroline Schnakers, Martin M. Monti

**Affiliations:** 1Department of Psychology, University of California Los Angeles, Los Angeles, CA 90095, USA; jpcoetzee@stanford.edu (J.P.C.); julia.crone@univie.ac.at (J.S.C.); micaceousjohnson@gmail.com (M.A.J.); lutkenhoff@g.ucla.edu (E.S.L.); 2Brain Injury Research Center (BIRC), Department of Neurosurgery, University of California Los Angeles, Los Angeles, CA 90095, USA; nspivak@mednet.ucla.edu (N.M.S.); creal@mednet.ucla.edu (C.R.); mblanco@mednet.ucla.edu (M.B.-B.); pvespa@mednet.ucla.edu (P.M.V.); 3UCLA-Caltech Medical Scientist Training Program, University of California Los Angeles, Los Angeles, CA 90095, USA; 4Department of Psychiatry, Stanford School of Medicine, Palo Alto, CA 94304, USA; 5Palo Alto VA Medical Center, VA Palo Alto Health Care System, Palo Alto, CA 94304, USA; 6Department of Neurosurgery, University of California Los Angeles, Los Angeles, CA 90095, USA; 7Research Institute, Casa Colina Hospital and Centers for Healthcare, Pomona, CA 91767, USA; cschnakers@casacolina.org

**Keywords:** acute, disorders of consciousness, ultrasound, subcortex, thalamus, deep brain stimulation, neuromodulation, non-invasive, treatment

## Abstract

The promotion of recovery in patients who have entered a disorder of consciousness (DOC; e.g., coma or vegetative states) following severe brain injury remains an enduring medical challenge despite an ever-growing scientific understanding of these conditions. Indeed, recent work has consistently implicated altered cortical modulation by deep brain structures (e.g., the thalamus and the basal ganglia) following brain damage in the arising of, and recovery from, DOCs. The (re)emergence of low-intensity focused ultrasound (LIFU) neuromodulation may provide a means to selectively modulate the activity of deep brain structures noninvasively for the study and treatment of DOCs. This technique is unique in its combination of relatively high spatial precision and noninvasive implementation. Given the consistent implication of the thalamus in DOCs and prior results inducing behavioral recovery through invasive thalamic stimulation, here we applied ultrasound to the central thalamus in 11 acute DOC patients, measured behavioral responsiveness before and after sonication, and applied functional MRI during sonication. With respect to behavioral responsiveness, we observed significant recovery in the week following thalamic LIFU compared with baseline. With respect to functional imaging, we found decreased BOLD signals in the frontal cortex and basal ganglia during LIFU compared with baseline. In addition, we also found a relationship between altered connectivity of the sonicated thalamus and the degree of recovery observed post-LIFU.

## 1. Introduction

Despite continued advances in life-sustaining intensive care for severe brain injury patients, little can be done to promote behavioral recovery in patients who fall into a coma, vegetative state (VS), or minimally-conscious state (MCS) (i.e., a disorder of consciousness; DOC) [1]. A general lack of clinical interventions persists, despite many recent advancements in the science of DOCs [2]. Such advancements include a growing emphasis on the role of deep-brain atrophy (e.g., in the thalamus and basal ganglia) in the impaired arousal and cognitive functioning common in DOCs [3]; however, clinical treatments which are able to target these nuclei safely in the DOC population are rare. Several emerging treatment options, some pharmacological [1] (e.g., amantadine and zolpidem) and some neuromodulatory (e.g., transcranial direct current stimulation [4]; tDCS, or thalamic deep brain stimulation [5]; DBS) ostensibly improve DOC symptoms by way of indirect (e.g., zolpidem) or direct (e.g., thalamic DBS) promotion of excitatory thalamic output to the cortex and, as a result, more neurotypical activity in cortico-basalganlgia-thalamo-cortical (i.e., mesocircuit) [3] and cortico-cortical [2,6] networks. To date, neurorestorative interventions are either systemic (e.g., pharmacological) or targeted (i.e., neuromodulatory technologies). With respect to the latter, important tradeoffs exist. Surgical techniques (e.g., DBS) possess the ability to target the deep nodes of the mesocircuit, with, at times, remarkable results, [5] at the cost of being applicable to only a small subset of patients [7] due to the risk [8] involved. However, the safety, ease, and broad applicability of well-established, non-invasive techniques (e.g., tDCS) are limited to reaching only the cortical nodes of the mesocircuit.

A renewed interest in low-intensity focused ultrasound (LIFU) as a method for obtaining spatially precise neuromodulation of deep brain structures without surgery may address this gap. Several experiments have now demonstrated the neuroactivity and safety of LIFU in both animal models [9,10] as well as in healthy human volunteers [11,12,13]. In addition, small case reports suggest the potential for this technique to produce clinically promising effects in both acute and chronic DOCs [14,15]. In what follows, we report the impact of magnetic-resonance (MR)-guided LIFU applied to the thalamus on brain activity and neurobehavioral measures in a convenience sample of acute DOC patients (*n* = 11). This work is part of the acute arm of a *first-in-man* proof-of-concept clinical trial (NCT02522429). While our results must be considered preliminary (as an uncontrolled phase 0 clinical trial), we report below three main findings: (i) significant behavioral improvements following LIFU, (ii) evidence of brain engagement during LIFU sonication, and (iii) a significant correlation between changes in connectivity in the thalamus targeted during LIFU and subsequent behavioral recovery.

## 2. Materials and Methods

### 2.1. Patients

This study included 11 acute DOC patients (see Table 1 for details). Patients were referred to the study following the persistence of DOC (as determined by coauthor P.V.) despite administration of routine first-line care at Ronald Reagan UCLA Medical Center after cessation of sedation protocols. An initial neurobehavioral evaluation with the JFK Coma Recovery Scale–Revised (CRS-R) [16] was conducted prior to enrollment to confirm eligibility (i.e., a persisting DOC).

Inclusion criteria were as follows:<6 weeks since injury;A Glasgow Coma Score <9 (at the time of injury);An abnormal CT;Prolonged loss of consciousness (>24 h);Behavioral profile consistent with a vs. or MCS, as assessed with the Coma Recovery Scale Revised.

Exclusion Criteria were as follows:Deep sedation;History of neurological illness prior to injury;Inability to safely enter the MR environment (e.g., ferromagnetic non-MR-safe implants).

### 2.2. Experimental Design

The overall experimental protocol is shown in Figure 1A. Briefly, patients underwent at least two, but more commonly three, baseline neurobehavioral assessments (at 1 week, 1 day and 1 h prior to LIFU; henceforth, pre-LIFU) followed by a session of LIFU, and two additional neurobehavioral assessments (at 1 h and 1 day following LIFU; henceforth, post-LIFU). Again, some patients also had a 1-week follow-up assessment. While the declared protocol called for a second, identical cycle of neurobehavioral assessments and LIFU, this was only possible for 27% of our sample, with the large majority of patients being discharged prior to undergoing the second session of LIFU. Finally, a follow-up assessment was conducted 1-week post-LIFU (from the first session, for patients who only underwent one LIFU, or from the second session, for the 3 patients receiving 2 LIFU sessions).

### 2.3. Neurobehavioral Assessments

Neurobehavioral assessments were conducted using the CRS-R [16]. Baseline responsiveness was assessed 1 week, 1 day, and 1 h prior to LIFU exposure, while responsiveness following the procedure was assessed 1 h, 1 day, and 1 week following LIFU exposure. Four patients, who were in the care of the Ronald Reagan UCLA Medical Center for longer periods, underwent the procedure twice, with 1 week separating each LIFU administration.

### 2.4. LIFU Sonication Protocol and Procedure

**LIFU Sonication Parameters.** In each session, LIFU was applied at 100 Hz pulse repetition frequency (PRF), 0.5 ms pulse width (PW), 650 kHz carrier wave frequency, 5% duty cycle (DC), and 14.39 $\frac{W}{{\mathrm{cm}}^{2}}$ I_SPPA.3_/719.73 $\frac{\mathrm{mW}}{{\mathrm{cm}}^{2}}$ I_SPTA.3._; “0.3” denotes tissue absorption at 0.3 dB/cm-MHz. This parameter set (PRF/PW/DC) has been derived from prior work demonstrating its neuroactivity [9,11]. Importantly, the energy levels employed in this experiment fall below the FDA limit for diagnostic ultrasound imaging of the human cranium [17]. The LIFU waveform was emitted from a single-element transducer (Brainsonix, Santa Monica, CA, USA; 71.5 mm curvature) positioned, using MR-guidance, such that its theoretical focus (55 mm from its surface) lay over the intended target. Once appropriate transducer placement was confirmed visually (see below), ultrasound was delivered inside the MRI for a total of ten 30 s blocks on, separated by 30 s off-periods (see Figure 1A).

**LIFU Target.** In light of the results from DBS applications to DOCs [5,7] and prior theoretical [3] and empirical [9,18,19] work, the intended LIFU target was the central thalamus. The protocol called for sonication to occur preferentially to the left thalamus, on the basis of prior work documenting a preferential association between atrophy in the left thalamus and the depth of the disorder of consciousness [18,19] Nonetheless, flexibility was allowed for patients with left craniectomy, which would result in higher than expected energy deposition into the target tissues, or left cranioplasty, given the unknown penetration and refraction profile of ultrasound through synthetic bone replacement materials. Additional flexibility was exercised in the case of implanted medical devices (e.g., stints or ventricular shunts) positioned proximally to the intended left hemispheric target and potentially susceptible to receiving significant energy deposition, and thus at risk of creating a potential hazard to the patient. (See Table 1 for laterality of LIFU administration in our sample.)

**LIFU Procedure.** The area surrounding the planned LIFU entry point on the head was shaved prior to positioning in order to minimize the impedance of ultrasound due to air bubbles. Ultrasound gel (aquasonic) was firstly applied to this region and smoothed in order to remove air pockets. The ultrasound transducer was then positioned so that its center lay on the squamous portion of the temporal bone (the thinnest part of the human skull) in order to minimize ultrasound scatter and refraction through the bone. A thin layer of gel was applied to the surface of the transducer, and bubbles were similarly smoothed from this layer. The transducer was then coupled to the head with gel filling any open space between the transducer membrane and the scalp with two straps—one horizontal and one vertical—securing the device to the patient. Conventional soft foam padding and pillows were used to further secure the positioning of the device and decrease the potential for head motion during the procedure. Next, we acquired a rapid (95 s) T1-weighted MPRAGE anatomical image (see 2.6). Using a circular MR fiducial and the visible center of the transducer, reference lines were drawn in the transverse and coronal planes, using the Siemens 3D display GUI available as part of the MRI device’s console software to visually locate the target of the LIFU beam in three dimensions. Adjustments to the positioning of the transducer on the head were made iteratively, re-acquiring a T1-weighted MPRAGE at each iteration, until the beam trajectory from the center of the transducer was assessed to be in-line with the intended target.

### 2.5. Behavioral Analysis

Behavioral data analysis was performed using JASP (JASP Team (2019). JASP, Version 0.11.1).

Behavioral responsiveness in patients was assessed using the total score on the CRS-R_index_ [20], which was calculated from the CRS-R using publically available scripts in R (Rstudio2021; https://github.com/Annen/CRS-R/blob/master/CRS-R_index.R, accessed on 8 January 2021). The CRS- R_index_ is a single value calculated from CRS-R subscores and was chosen because it is thought to more appropriately represent functional recovery with a single number. Prior to analysis, the highest CRS-R_index_ score for each experimental period (i.e., Pre-LIFU 1, Post-LIFU 1, and, for patients who had a second session, Post-LIFU 2) was taken in order to best capture patients’ maximal performance. For patients who had 2 runs (*n* = 3), recovery following LIFU 1 and that following LIFU 2 were averaged for inclusion in group-wide statistics. However, behavioral analyses were also performed following the exclusion of run 2 for these patients and these results are also reported below. Given that data was found to be non-normal (Shapiro-Wilk), a non-parametric Wilcoxon signed rank test was used to compare Pre-LIFU and Post-LIFU scores for all patients.

### 2.6. MRI Sequences and Scanning Procedure

Prior to transducer placement and targeting, a high-resolution T1-weighted MPRAGE (TR = 2.08, TE = 4.14, voxel size = 1 × 0.5 × 0.5 mm^3^) was acquired for later processing. Rapid (95 s) T1-weighted structural sequences (TR = 1900 ms, TE = 2.2 ms, voxel size 2 mm^3^) were used for targeting purposes. Concurrent with LIFU administration, BOLD data was collected with a T2*-weighted Echo Planar Image sequence (TR = 2 s, TE = 25 ms, voxel size = 3.44 × 3.44 × 4.25 mm^3^).

### 2.7. MRI Data Analysis: Preprocessing

MRI data preprocessing and analysis were conducted using FSL (FMRIB Software Library v6.0.1) [20] with in-house Bash shell scripts. In addition, second-level data analysis was performed using JASP. JASP Team (2019). JASP (Version 0.11.1).

In order to produce group-level functional results, data from patients who received LIFU to the right thalamus were flipped such that the right hemisphere became the left hemisphere. This includes structural data for the purpose of co-registration. Next, preprocessing was performed including brain extraction (using optiBET [21], given its superiority in DOC patient data), spatial smoothing (using a Gaussian kernel of 5 mm full-width half-max), slice timing correction (Fourier-space time-series phase-shifting), highpass temporal filtering (Gaussian-weighted) at 0.01 Hz, and motion correction (MCFLIRT) [20,22]. With the exception of brain extraction, these procedures were performed in fsl FEAT.

Following recent data [23], in-scanner head motion was mitigated by including in the statistical model a number of nuisance regressors, including individual time points with excessive motion (i.e., spike regression [23]) derived from the output of fsl FLIRT, 24 head-motion parameters, and regressors for white matter and CSF components. White matter and CSF regressors were produced by segmenting T1 images for each patient using FSL Fast (visually inspected for accuracy). Tissue segmentations for white matter and CSF were moved into functional space (for some patients, FSL epi_reg was employed while some patients required nonlinear registration using FSL FNIRT) and binarized (and again visually inspected for accuracy). Time series for white matter and CSF were then extracted from functional images using fslmeants. Framewise displacements for each volume were derived from FSL MCFLIRT [20,22], and these were used to exclude unwanted volumes with a framewise displacement exceeding 0.5 mm (25% of voxel width). Any functional data not exceeding 4 min of total time [23] were excluded (a single functional run, specifically patient 4, run 2) from the rest of data analysis, while no patients were dropped.

In order to register structural images to functional space, we employed a combination of FSL epi_reg, which is tailored for the coregistration of subcortical regions in particular (including the LIFU target) and conventional 12 dof linear coregistration (using FSL FLIRT). Both were run for each registration and visually assessed for which was more successful. In order to register structural images to standard space, nonlinear registration (FSL FNIRT) was used. For some patients, however, linear registration (FSL FLIRT) resulted in better alignment of subject’s structural images to the standard MNI template space, as determined by visual inspection conducted prior to analysis.

### 2.8. BOLD Data Analysis: Effect of LIFU on Activity and Behavior

BOLD data collected during LIFU were first analyzed employing a univariate general linear model (GLM) approach, [24] including pre-whitening correction for autocorrelation (FILM). A univariate analysis was conducted using a single “task” regressor, which represented the onset time of 30 s blocks of LIFU administration. Thus, here the “baseline” conditions used were the inter-sonication periods where no LIFU was applied. For each BOLD sequence, we computed 2 contrasts: LIFU > no LIFU and LIFU < no LIFU, and assessed each using a fixed effects model given the low sample size. For patients with two LIFU exposures, results from the two runs were averaged at level two prior to third-level fixed-effects analysis. At the third level, data were cluster corrected for multiple comparisons using a cluster-level threshold of z > 3.09 (corrected *p* < 0.05). A separate level 3 analysis was conducted with cluster correction at z > 2.57 (corrected *p* < 0.05) [25]. Z-scores of 3.09 and 2.57 correspond to *p*-values of 0.001 and 0.05, respectively.

In order to determine if the degree of LIFU-induced modulation was associated with subsequent neurobehavioral change, an additional regressor was included in the third-level (group) analysis capturing each subject’s recovery post-LIFU and measured using the CRS-R_index_.

### 2.9. BOLD Data Analysis: Effect of LIFU on Connectivity and Behavior

In order to determine whether the connectivity of the thalamus was modulated by LIFU sonication, we performed a psychophysiological interaction (PPI) analysis, a technique designed to detect changes in connectivity between a seed region and the rest of the brain as a function of the onset and offset of a psychological task [26]. In this case, we were interested in changes in connectivity which occurred as a function of the onset and offset of LIFU and so our “psychological” regressor reflects those. The thalamic seed was obtained by performing subject-specific segmentations obtained from each patient’s high-resolution T1-weighted image using FSL FIRST. All segmentations were visually inspected for accuracy prior to conducting the analysis. The time series of thalamic BOLD was extracted from each functional run using a mask for each thalamus with fslmeants. The PPI was estimated for each patient separately and aggregated at the group level with the same procedure as outlined above for the full-brain analysis in Section 2.8.

In order to determine if the results of this PPI analysis covaried with behavioral recovery, we included in the group analysis a regressor describing each subject’s behavioral change post-LIFU as measured using the CRS-R_index_.

### 2.10. Thalamic ROI Effect

In order to determine if a change in the BOLD signal was observed in the thalamus itself during LIFU, thalamic ROIs created using fslmeants (as explained in more detail above) were used to extract the mean z-score representing the change in the BOLD signal observed during LIFU within each thalamus, within each run, and within each patient. Again, for those subjects who experienced two runs, values were averaged. These z-scores were then compared, in a two-tailed, one-sample *t*-test, to a value of 0 (no significant effect). Moreover, the targeted and non-targeted thalami were also compared using a two-tailed within-subjects *t*-test.

### 2.11. Safety Measaures

With respect to safety, we recorded vital parameters during LIFU administration and MR data collection (e.g., heart rate, blood oxygen, and blood pressure). Furthermore, any adverse events that occurred in patients over the course of the study were recorded.

## 3. Results

### 3.1. Behavioral Analysis

We found a significant increase in maximal responsiveness (i.e., highest CRS-R_index_ score) among patients (*p* = 0.014; see Figure 2) following the LIFU procedure compared to baseline. The analysis was significant also when repeated on the raw CRS-R total score (*p* = 0.014). The finding was unchanged when analyzing only the data from the first LIFU session for all patients (i.e., when excluding the data from the second session administered in 3 patients only; *p* = 0.009 and *p* = 0.008 for the CRS-R_index_ and CRS-R total score, respectively). However, when comparing CRS-R_index_ scores immediately prior to and immediately following LIFU administration—which better reflects the immediate response to LIFU—no significant change was found (*p* = 0.820). See discussion for how these results may reflect either the true time-varying effects of LIFU or fatigue induced by the procedure. Furthermore, behavioral recovery was found to positively correlate with initial CRS-R_index_ scores (highest taken prior to LIFU; Spearman’s Rho = 0.651, *p* = 0.015). For a complete listing of patient behavioral data, see Appendix A.

### 3.2. BOLD Data Analysis: Effect of LIFU on Activity and Behavior

As compared with baseline (i.e., LIFU-off), 30 s of deep-brain LIFU sonication resulted in significantly reduced BOLD signals in three anterior clusters (see Figure 3A). Specifically, these clusters subsumed portions of the subcallosal prefrontal cortex, anterior cingulate cortex, medial prefrontal cortex, and striatum (both caudate and putamen; ipsilateral to the sonication site). None of these activations appeared to correlate with subsequent behavioral recovery (see Figure 3B).

### 3.3. BOLD Data Analysis: Effect of LIFU on Connectivity and Behavior

Our psychophysiological interaction analysis [26] (Figure 3C) found that during LIFU sonication the targeted thalamus increased its connectivity with two clusters—one in the ipsilateral pre- and post-central gyrus and one subsuming portions of the contralateral opercular and insular cortex—while decreasing its connectivity with the ipsilateral frontal polar cortex (see Figure 3C). When the same analysis was run on the thalamus contralateral to sonication, no significant change in connectivity was observed during LIFU (see Figure 3E).

We also found that this PPI effect was predicted by behavioral recovery (Figure 3D). Specifically, we found that decreased connectivity between the targeted thalamus and regions in the frontal lobe, spanning bilaterally the dorsal and medial frontal cortices, bilateral insula, and bilateral subcortical structures, was associated with increased behavioral responsiveness following LIFU sonication. Specifically, significant regions included portions of the ipsilateral, dorsolateral prefrontal, and motor cortices, the bilateral striatum, the contralateral globus pallidus, contralateral thalamus, the contralateral opercular cortex, the subcallosal frontal cortex, the anterior cingulate cortex, and bilateral orbitofrontal cortex. Portions of the clusters over the contralateral basal ganglia structures, opercular cortex, amygdala, and anterior cingulate cortex retain significance when using a more conservative CDT of 0.001 and both thresholds are shown in Figure 3D.

Furthermore, we found that increased connectivity between the targeted thalamus and regions throughout the contralateral motor cortex, the parietal and temporal lobes, and the occipital cortex was also associated with increased recovery in patients (Figure 3D). Specifically, these regions included portions of the somatomotor cortex, the middle temporal gyrus, the occipital pole, and the precuneus. Portions of the clusters found in the occipital pole and somatomotor cortex retain significance when using a more conservative cluster determining threshold (CDT) of 0.001 and both thresholds are shown in Figure 3D.

When the same analysis was run using the thalamus contralateral to the LIFU sonication as a seed, no significant results were found (Figure 3F).

### 3.4. Thalamic ROI Effect

An ROI analysis of the targeted thalamus reveals that while the BOLD signal was numerically lower during sonication (as compared to baseline), the change was only at trend level (*p* = 0.097). Nonetheless, the change was significant when compared with the non-targeted thalamus (*p* = 0.047).

### 3.5. Safety of Thalamic LIFU

In regard to safety, no changes in vital parameters (e.g., heart rate, blood pressure, or oxygen concentration) were observed during the administration of LIFU. While two adverse events (AE) occurred in patients during the study, both were considered unrelated to the LIFU procedure. One AE involved respiratory suppression of a patient prior to any LIFU exposure, while the other resulted from a seizure in one patient more than one week after LIFU in the context of sepsis.

## 4. Discussion

Firstly, with respect to feasibility and safety, no adverse events associated with the application of LIFU were observed over the course of this study; thus, our results support the apparent safety of thalamic LIFU in acute DOCs at the parameters tested, which is in line with the known safety profile of transcranial ultrasound [17,27]. Moreover, our findings suggest that MR-guided LIFU can be accomplished in acute DOC patients while viable functional data are collected despite the challenges that equipment placement and patient motion present to this procedure.

Secondly, with respect to behavior, this cohort increased in their neurobehavioral responsiveness following thalamic LIFU, in line with some prior case reports in acute [15] and chronic patients [14]. Specifically, this reflects an increase in the highest CRS-R_index_ score in the one-week period following LIFU when compared to the best CRS-R_index_ score at baseline. In four of eleven patients, this included a shift up in diagnostic category (e.g., vs. to MCS). This improvement correlated positively with the initial level of patient functioning, suggesting that the mechanism of this recovery may require some minimal level of neurotypicality. However, this early finding should not deter future investigations in lower-functioning patients, which may confirm or dispel such an observation. Indeed, even some vs. patients enrolled in this study demonstrated apparent recovery. Interestingly, no significant difference was found when comparing the CRS-R_index_ immediately preceding and immediately following LIFU application. While this null result cannot bolster or dispel the notion of rapid recovery, our behavioral results, when taken as a whole, suggest that recovery—if indeed induced by thalamic LIFU—may require some time after the 1 h post-LIFU period to develop. However, a major confound here is that the lengthy procedure (MR Imaging and two CRS-R assessments in one day) is likely to induce fatigue in patients, which may mask any immediate effect.

Finally, our functional MRI results provide initial data on the neural origin of this apparent behavioral effect. The results of our block design model suggest that an acute reduction in activity, instead of acute excitation, is induced by thalamic LIFU when compared with baseline. Portions of both the anterior cingulate, subcallosal, and medial prefrontal cortex appear inhibited during LIFU-on blocks. Furthermore, the ipsilateral striatum (both caudate and putamen) was inhibited during LIFU-on blocks. While no thalamic cluster appeared in the whole-brain results, the sonicated thalamus had a reduced BOLD signal compared to the un-sonicated thalamus during LIFU blocks in an ROI approach. This pattern of results is interesting when considering the intimate connectivity known to exist between the cortex (especially the frontoparietal), the basal ganglia, and the targeted central thalamic regions [3,28]. Whole-brain regions of reduced BOLD signals with small effects in the targeted nuclei mirror results found in a previous LIFU study that targeted the thalamus and adjacent basal ganglia in healthy individuals using the same parameter set [11]. Moreover, an observation of acute inhibition is in line with recent associations between low duty cycle (here 5%) in LIFU parameter sets and inhibition [29]. As inhibition from LIFU is thought to involve the excitation of inhibitory neurons (cortical interneurons or thalamic reticular cells), a local BOLD effect—driven largely by local glutamate secretion and metabolic changes [30]—may be difficult to detect [29,31].

Our psychophysiological interaction (PPI) results suggest a more complex change in connectivity between the targeted thalamus and the rest of the brain when LIFU is applied. During LIFU-on blocks, the targeted thalamus decreased its connectivity with the ipsilateral fronto-polar cortex. However, it increased its connectivity with the ipsilateral somatomotor cortex and contralateral opercular/insular cortex. Perhaps more interestingly, changes in thalamic connectivity which predicted recovery were more expansive and generally aligned with changes in BOLD signals during LIFU. Indeed, reduced connectivity between the targeted thalamus and all of the regions which we found to be inhibited during LIFU-on blocks were associated with greater recovery (i.e., anterior cingulate, subcallosal frontal cortex, medial prefrontal cortex, and ipsilateral striatum). However, the effect expanded to include portions of the ipsilateral prefrontal cortex, contralateral striatum, bilateral opercular cortex, and the contralateral thalamus as well. Furthermore, increased connectivity between the targeted thalamus and large portions of the contralateral parietal and occipital lobes and the motor cortex also predicted recovery. It is interesting that this increase in connectivity was entirely contralateral; one hypothesis is that this may reflect a form of compensation for thalamocortical connectivity changes that were induced ipsilaterally.

Strikingly, no significant changes in connectivity were found between the non-targeted thalamus and the rest of the brain, nor did recovery predict changes in connectivity.

### 4.1. The Potential Benefits of Inhibition

Given that DOCs are often associated with a gross reduction in neural activity [32] compared with that of healthy patients, it may appear counterintuitive that we observed behavioral recovery following apparent inhibition. While reduced activity in large-scale cortico-subcortical networks is a hallmark of the DOC pathology [3], a more complete description of the neural correlates of DOCs may instead emphasize a more general dysregulation of large-scale networks and the isolation of independent regions [2]. It is relevant to note here that some DOC patients present with normative levels of whole-brain metabolism [32]—their condition instead being thought to result from functional network changes instead of reduced whole-brain arousal—even in regions distant from the site of injury (see diaschisis) [33].

From this perspective, interventions which cause acute excitation as well as inhibition may restore more neurotypical states dormant within highly damaged brains [34]. Indeed, CNS depressants (e.g., zolpidem, baclofen, lamotrigine, and lorazepam) have been associated with recovery in select DOC patients [35,36]. Although the mechanisms behind these effects remain debated, CNS depressants can, even in healthy brains, increase functional connectivity [37]. As has been previously proposed [34], inducing inhibition within the brains of DOC patients may induce recovery by bolstering the inhibitory gating mechanisms necessary to conduct the large-scale connectivity presumed to underlie goal-directed activity. It could be argued that this perspective is especially relevant to thalamic modulation as the role of this structure in cognition appears to rely greatly on sensorimotor gating [28,38,39]. Once re-established, improved functional connectivity may evolve or become self-sustaining [3], which may explain reports of the CNS depressants baclofen and lamotrigine being associated with improved symptoms in DOC patients weeks after administration and in a pattern unrelated to these drugs’ pharmacodynamic profiles [34]. This may similarly explain why we do not observe recovery in our patients in the immediate post-LIFU assessment but only after a period of time has passed.

In the perspective that perturbation of not only brain activity, but especially brain connectivity, is important for recovery from DOCs, it is exciting that we found a complex pattern of altered connectivity with the targeted thalamus during LIFU that was related to behavioral recovery. Contrarily, the reduction in the BOLD signal found during LIFU did not predict recovery. Based on these results, we may hypothesize that acute perturbation of thalamic connectivity induced by thalamic LIFU may have a beneficial effect on restoring the more normative patterns necessary for behavioral recovery. However, this should remain a tentative hypothesis awaiting more extensive future investigations.

### 4.2. Limitations and Future Directions

Though we found some trend-level evidence for the inhibition of the targeted thalamus as a whole during LIFU, such an approach neglects the thalamus’ complexity—its many nuclei, their possible interactions (mediated by the thalamic reticular nucleus), and subtypes of thalamic neurons. Recent computational models suggest that LIFU applied at a 5% duty cycle preferentially causes action potentials in excitatory thalamocortical cells compared to inhibitory thalamic reticular cells (RE); however, raising the DC to just 7% produces equal action potentials between both thalamic cell subtypes [29]. Thus, passing a beam of LIFU at a DC of 5%, which is close to the critical threshold of RE neuron activity, through a large portion of the residual thalamus likely has a complex effect on individual thalamocortical circuit relationships. In turn, these relationships likely depend strongly on the precise shape of local connectivity. This complexity may be mirrored in our connectivity results, which showed a pattern of both increases and decreases in thalamic connectivity that could reflect differences in the local thalamic effect of LIFU or cortical target cell types.

Complexity is further added by any spatial imprecision associated with LIFU applied using single-element transducers. The known accuracy of LIFU emitted from a single element transducer, though far greater than other noninvasive techniques, is relatively lower than the theoretical precision of DBS, the use of which in DOCs greatly inspired this work [40]. However, it should be considered that highly precise DBS electrode placement is not trivial and, when considered, can arguably create room for error more comparable to that of LIFU [41,42]. While the focus of the transducer used here has been measured to extend roughly 0.5 cm laterally and 1.5 cm longitudinally in water [11], the perturbation of this focal shape by the skull is likely to add an additional ~1 cm (a rough estimate, erring on the higher end) of possible deviation in any direction [11,32] and generally expand the focal area laterally [11]. While we did use MRI guidance to precisely target the central thalamus, we did not use individualized modeling approaches to account for these skull-refractory effects, which were expected to be in line with those previously reported in healthy humans since care was taken to avoid passing energy through damaged or displaced bone. If the goal of this study was to selectively target the central lateral (CL) nucleus of the thalamus, a small nucleus often targeted with DBS [5], this degree of imprecision was perhaps inappropriate. However, the greater central thalamus has been robustly associated with arousal regulation and was considered an appropriate target here. Indeed, previous use of DBS in DOC patients has acknowledged the likely coactivation of nearby central thalamic regions [40] (e.g., the paralaminar regions of the median dorsalis, and the posterior-medial aspect of the centromedian/parafascicularis nucleus complex). Concentration of energy on the central thalamus—avoiding much energy deposition in, e.g., the anterior or pulvinar thalamus—is possible with the known accuracy of the LIFU procedure used here. However, the larger longitudinal extent of the focus likely resulted in direct impacts on basal ganglia structures nearby the central thalamic target (e.g., the globus pallidus interna), which can be considered a limitation of this study. See Cain et al., 2021 [11], for a more detailed discussion of the effect of the skull on the transducer and sonication parameters used here. Some of these challenges could be avoided in the future by employing functional localizers and more advanced structural imaging techniques [43] to locate thalamic subregions in damaged brains alongside the use of more spatially-precise multi-transducer arrays.

Furthermore, many of the ambiguities that remain could be alleviated by larger datasets with more numerous time points and the addition of a control condition. For instance, we could not probe the effect of LIFU on individual sub-scores of the CRS-R_index_ (e.g., arousal or auditory components) due to insufficient power. Moreover, greater statistical power (e.g., due to a larger sample) would allow us to probe the possible influence of patient comorbidities (e.g., by including them as covariates in the statistical models), which may influence patient responsiveness independent of LIFU. Future analyses of this kind may reveal the mechanisms behind the general recovery observed here—whether it be mediated by improved arousal or increased complex command following, which each imply different mechanistic underpinnings. Similarly, behavioral and neuroimaging data with more time points could map the path of recovery following LIFU and provide a richer dataset from which to probe the network interactions underlying that recovery. Future investigations may opt for jittered event-related designs when collecting neuroimaging data; our contrast of LIFU-on vs. LIFU-off blocks may be unable to disambiguate acute effects from regularly lagged (i.e., a lag of ~30 s) rebound effects from LIFU administration. Thus, the findings reported here concerning the valence of thalamic LIFU’s influence on brain activity in DOCs should be confirmed by more complex designs.

A major limitation of the present work, typical of a proof-of-concept trial, is the absence of a sham-control group. While these patients were enrolled in our trial specifically because they were not showing spontaneous recovery, it is not impossible that they spontaneously recovered function in the week following LIFU, resulting from only the passage of time. Furthermore, a substantial degree of variability in responsiveness is to be expected in patients in the acute phase of DOCs, especially when assessed in an ICU setting (e.g., due to changing ICU rooms, differential noise levels, unstable medical conditions, or changing administration of medications). However, it is worth noting that the CRS-R has been validated in acute DOC patients despite these challenges which plague all behavioral assessments. Moreover, assessment of all patients began after they were considered medically stable and off sedatives (at least 5 days after injury). Yet, while the change in behavioral responsiveness after sonication needs to be interpreted with caution, it is particularly noteworthy that the degree to which LIFU modulated the connectivity of the thalamus with the fronto-parietal and subcortical areas is associated with the behavioral recovery observed following LIFU. Of course, whether this indicates a positive effect of LIFU on brain dynamics—and, subsequently, behavioral responsiveness—or whether we are detecting, through thalamic responsiveness to LIFU, patients with sufficiently preserved thalamo-cortical connectivity—such that they are more likely to recover—remains to be assessed.

## 5. Conclusions

We found preliminary evidence for the safety and feasibility of LIFU in acute DOCs. In addition, we found that the degree to which LIFU could alter thalamic connectivity was associated with subsequent behavioral recovery. Functional data collected during LIFU administration suggest that the acute effect of LIFU may be inhibitory at these parameters, in line with prior investigations. However, these data emphasize the role of changes in connectivity (both increases and decreases) with the thalamic target in the behavioral recovery of patients. Future investigations would benefit greatly by including neuroimaging at more time points to better parse the neural underpinnings of apparent recovery from thalamic LIFU, especially considering that this recovery appears to develop over time. While many unknowns remain, these preliminary results should help compel and define future efforts to assess the efficacy of thalamic LIFU as a treatment for DOCs and its associated mechanisms.

## Figures and Tables

**Figure 1 brainsci-12-00428-f001:**
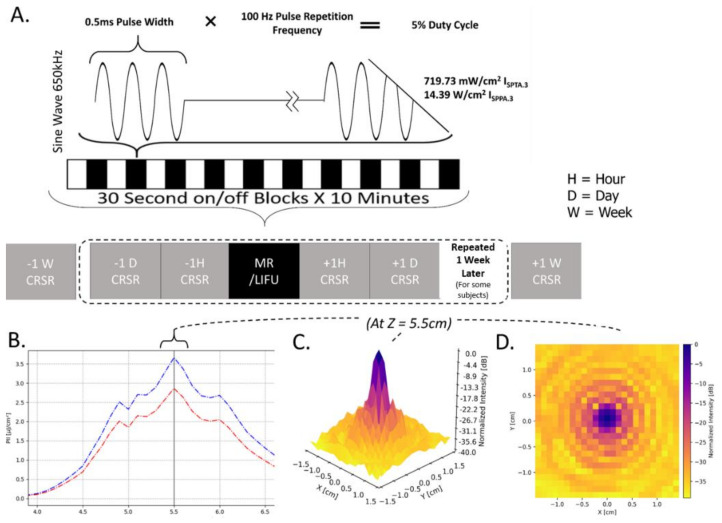
(**A**) Depiction of study protocol involving LIFU parameters and CRS-R (Coma Recovery Scale-Revised) assessments. ISPTA.3 = Spatial Peak Temporal Average Intensity.3. ISPPA.3 = Spatial Peak Pulse Average Intensity.3 (“0.3” denotes deration (attenuation) due to absorption by tissue at 0.3 dB/cm-MHz). (**B**) Intensity in the longitudinal plane (Z plane, extending from transducer) in absolute (pulse intensity integral (PII); “0.3” denoting absorption in human tissue at 0.3 dB/cm-MHz) values of Z correspond to distance from the transducer surface. Note the peak intensity 5.5 cm from the transducer surface and that a 50% (−3 dB) reduction in peak intensity is found in an area approximately 1.5 cm in length. (**C**,**D**) Intensity in the radial plane (X/Y plane, extending from focal point of ultrasound beam 5.5 cm from transducer surface) shown in both 3 (**C**) and 2 (**D**) dimensions. A 50% (−3 dB) reduction in peak intensity occurs in an area approximately 0.5 cm in width. Note that the decibel scale is nonlinear and −3 dB approximately corresponds to a 50% reduction in intensity; this scale is normalized to maximal intensity, where peak intensity equals 0 dB.

**Figure 2 brainsci-12-00428-f002:**
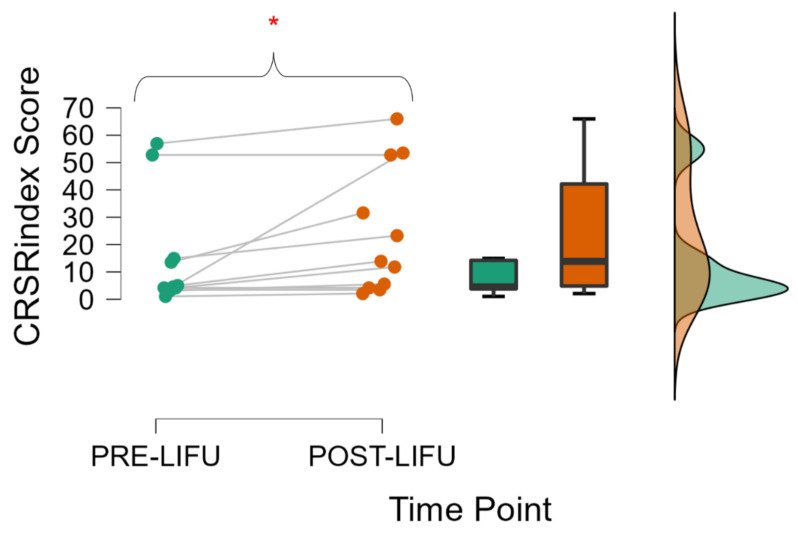
Points, box plots, and distribution of the highest CRS-R_index_ score prior to and up to one week following LIFU. Dashed lines represent the median of each distribution. The red asterisk indicates a significant difference between pre- and post-LIFU scores.

**Figure 3 brainsci-12-00428-f003:**
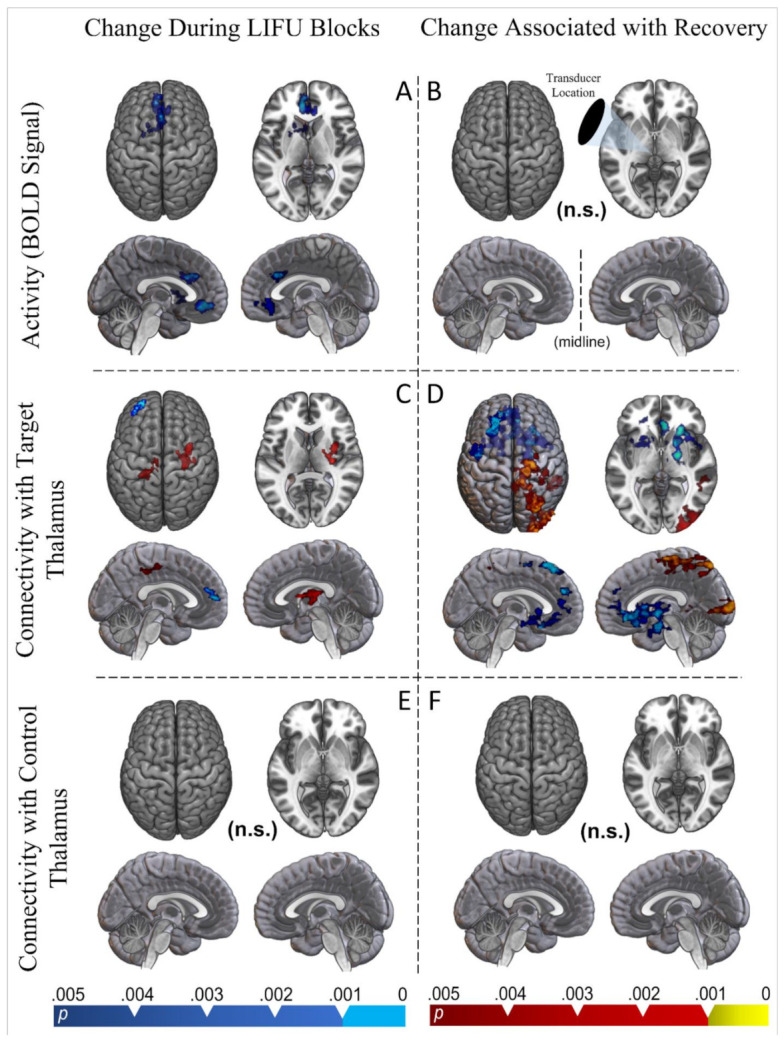
Whole brain results. For all analyses, statistical maps were obtained using a fixed-effects model as implemented in FSL6.0.1, and are shown at two levels of cluster correction for multiplicity (CDT set at *p* < 0.005, in blue, and at *p* < 0.001 in violet). Note that sagittal images are shown from the midline and each shows one hemisphere. (**A**) Regions of significant change in BOLD signals during sonication compared to inter-sonication periods (i.e., baseline). (**B**) Regions of significant BOLD signal change predicted by behavioral recovery. (**C**) Connectivity changes observed during LIFU-on blocks compared to LIFU-off blocks (PPI) between the whole brain and the targeted thalamus. (**D**) These changes were predicted by behavioral recovery. (**E**) Connectivity changes observed during LIFU-on blocks compared to LIFU-off blocks (PPI) between the whole brain and the non-targeted (control) thalamus. (**F**) These changes were predicted by behavioral recovery.

**Table 1 brainsci-12-00428-t001:** Relevant patient-specific information.

Patient #	Age	Sex	TSI	Etiology	Initial	Final	LIFU #	Hem.
1	25	M	18 d	TBI/MVA	MCS+	MCS+	1	R
2	23	F	16 d	TBI/MVA	MCS−	MCS−	1	R
3	45	M	5 d	TBI	MCS+	EMCS	2	L
4	72	M	5 d	Glioma/Stroke	coma	VS	2	L
5	75	F	28 d	TBI	VS	MCS−	1	L
6	59	M	17 d	TBI	VS	MCS+	1	L
7	67	M	4 m	TBI/MVA	MCS+	MCS+	2	L
8	22	M	13 d	TBI	coma	coma	2	R
9	31	M	1 m	TBI/MVA	MCS+	MCS+	1	L
10	53	M	1 m	TBI/MVA	MCS+	MCS+	1	L
11	31	M	24 d	TBI/MVA	VS	VS	1	L

TSI = time since injury. TBI = traumatic brain injury. MVA = motor vehicle accident. Initial diagnostic category (e.g., MCS+) is based on the highest performance measured using the CRS-R prior to LIFU exposure. Final diagnostic category is based on the highest performance measured using the CRS-R in the week following the final LIFU exposure (regardless if 1 or 2 sonications were performed). The values of total CRS-R scores from which these diagnostic categories are derived are also provided in the CRS-R PRE and POST columns. The number of sonications (LIFU #) and the hemisphere of the targeted thalamus (Hem.) are reported.

## Data Availability

The data presented in this study can be obtained through a Material Transfer Agreement with UCLA.

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
