# Peer review of "Ultrasonic Deep Brain Neuromodulation in Acute Disorders of Consciousness: A Proof-of-Concept"

_brainsci, 2022, doi:10.3390/brainsci12040428_

Round 1
Reviewer 1 Report
The lack of therapeutical options for DOC patients makes this article, although based on preliminary data, absolutely relevant.
Strengths of the article:
1) the safety/feasibility of LIFU
2) the measurable fMRI effect (maybe some complementary methods such as TMS-EEG should be discussed as future directions)
I have however some observations that the authors :
1) the cohort of patients, treated in an early phase, without sham/control does not allow to draw conclusions on clinical efficacy. The observed behavioural changes might be just a consequence of the normal evolution in the early phase after a brain injury (i.e. time-effect); this point should be emphasised and the presentation of the behavioural changes should be more critical/prudential
2) the figure 2 should be substituted with a Table with all the clinical-behavioural data of all the patients
3) CRS-R has several limitations when used in an ICU setting/early phase, this should be discussed
Reviewer 2 Report
In this paper, Cain et al. aim at assessing the effects of thalamic LIFU on behavioral recovery and brain activity of DOC patients using resting state MRI bold signal as surrogate marker.
Briefly, 11 patients were included in this study. All had brain injury (within 6 weeks according to inclusion criteria) of different causes but mainly after head trauma.
Patients were assessed at baseline (1 week, 1 day and 1 hour) prior to LIFU session then 1 hour, 1 day and 1 week after LIFU. Four patients had 2 LIFU sessions. Eight patients were treated on the left thalamus whereas the three remaining received LIFU on the right side for technical reasons.
The main results are
- Improvement on the maximal CRS-Rindex after LIFU vs baseline but no difference when immediately after- and immediately prior- CRS-Rindex were compared. Some patients changed their diagnostic category.
- Brain engagement during LIFU session with changes in BOLD signal and thalamic connectivity (with both increase and decreased signal)
- Functional recovery was predicted by i) decreased connectivity between sonicated thalamus and bilateral frontal and sub-cortical regions : ii) increased connectivity between targeted thalamus and contralateral motor, parietal, temporal and occipital cortex.
Conceptually, this is a very interesting work. Neuromodulation of the thalamus seems to be a promising approach to treat patients with DOC and LIFU offers a non-invasive well tolerated mean to reach this objective.
However, I have several major concerns that in my opinion, prevent to draw any firm conclusion from the present work both regarding the behavioral and MRI effects:
- As stated by the authors in the discussion, there is no control group and it is really not possible to rule out that the clinical improvement is simply due to the “natural” evolution after the brain injury and unrelated to the LIFU procedure. Longer baseline period with more repeated assessments or sham sonications could have allowed to overcome this limitation. In addition, CRS-R could not be assessed blindly to LIFU condition. This could lead to a major assessment bias. Thus, it is difficult to be sure that there is an actual clinical benefit related to LIFU procedure.
- I have also some concerns regarding the way thalamus is targeted. There is a single theoretical focus (55 mm from the transducer) that is not “corrected” for the bone thickness / conformation of each patient. I wonder If previous experimental work have demonstrated that the brain volume targeted in such conditions is reproducible across patients. In addition, when thalamus DBS was used for DOC (see Schiff et al. Nature 2007), the leads were implanted in order to have each contact within the central lateral nucleus, paralaminar regions of the median dorsalis, and the posterior-medial aspect of the centromedian/parafascicularis nucleus complex). The assumption was that these thalamic regions were strongly involved in cortical activation. In the present work, the targeting is clearly not precise and I really wonder what is the actual volume of tissue “modulated” by LIFU. Could the authors comment on this point? Also, surrounding structures may be involved in the modulation “field”.
- The MRI were performed during sonication. Surprisingly, the authors observed that the clinical effect was predicted by decreased connectivity of the targeted thalamus with bilateral frontal and subcortical regions. This contra-intuitive result was discussed and the authors propose a seducing hypothesis regarding the potential benefit of “inhibition”. This could be an acute effect of LIFU and serial MRI scan at different time points may have been helpful to get a better picture of the mechanisms underlying a potential effect.
Minor points
- Some sentences in the introduction (line 39-43 for example) are difficult to follow
- There are conflicting statements regarding
- The number of patients that had 2 sessions of LIFU (3 according to the 2.2 experimental design section and 4 according to the table and section 2.3). Could the authors comment on this point ?
- The number of patients that changed their diagnostic category (3 according to the table and 4 according to the discussion, page 11 line 365).
- For one patient (patient 7) , the delay from brain injury was 4 months whereas according to inclusion criteria (see page 2 line 79), the delay since injury should be < 6 weeks.
Reviewer 3 Report
The article outlines benefits of the new method of non-invasive neuromodulation approach with ultrasound in patients with DOC. As recovery of consciousness still remains a huge non-solved problem, the appearance of new methods may be very useful further.
The methodology and design seems correct for the research of DOC. Nevertheless, there are some uncertainty that should be mentioned.
First of all, the number of participants is too low to count significant statistics. Also the absence of sham group making the results more approved. Thus, the studies with DOC patients are always poor with numbers because of low occurrence of these patients. Should be mentioned, that the absence of sham group defined as the most important limitation of the study by authors in the last paragraph of discussion.
The second problem for my opinion is that included patients were in acute period of their disease. That fact can affect directly on the results - authors could observe normal evaluation of TBI with increasing consciousness, and the role of the LIFU in that process may be overestimated. This fact discussed in the limitations also.
But on the other hand, acute state could be accompanied with the average of comorbidity that could affect the level of consciousness as well, but in the negative way. It is mentioned that at least one patient had sepsis, so could had others, too. In that point, the final assessment with CRS-R and detecting of diagnosis of DOC can be not correct enough, because consciousness could be covered by poor somatic status. In this state increasing in CRS-R may be connected to improving of comorbidities but not to LIFU. And also, we can loose the progress after LIFU because of severe patient’s condition. It would be interesting to make some analysis depends on comorbidities in the group or at least mentioning about facts like sepsis or other somatic complications, that could affect consciousness as well.
All listed problems above show the need of control group in studies like that. In acute DOC it is more feasible. Nevertheless, it could be the point of further investigations of the authors.
In addition, for me would be interesting to see concrete CRS-R scores for patients before-after near the DOC diagnosis (Fig. 1A). It seems that the table will not be overloaded if this numbers will be added.
It will be interesting also to see longterm results, if any patient recovered consciousness and with what deficit.
In conclusion, the article is very interesting and important for today neuroscience. Firstly because we still don’t have approved methods for consciousness recovery. Secondly, because it has a huge block of functional MRI investigation before and after LIFU was applied. In this point, the most valuable result here is about neuroimaging, that shows, that LIFU affects on pathways connected to target thalamus.
I think, the article is informative, new and useful, so I suggest to publish it after minor revisions, if authors would consider it possible.
Round 2
Reviewer 1 Report
The article is now more balanced and deserves publication
Reviewer 2 Report
The authors have adressed my concerns